

# Winter temperature predicts prolonged diapause in pine processionary moth species across their geographic range

Md H.R. Salman[1], Carmelo P. Bonsignore[2], Ahmed El Alaoui El Fels[3], Folco Giomi[1], José A. Hodar[4], Mathieu Laparie[5], Lorenzo Marini[1], Cécile Merel[1], Myron P. Zalucki[6], Mohamed Zamoum[7] and Andrea Battisti[1]

[1] DAFNAE, University of Padova, Legnaro Padova, Italia
[2] PAU, Università Mediterranea, Reggio Calabria, Italia
[3] Faculté des Sciences, Université Cadi Ayyad, Marrakech, Morocco
[4] Department of Ecology, Universidad de Granada, Granada, Spain
[5] Unité de Recherche de Zoologie Forestière, INRA, Orleans, France
[6] School of Biological Sciences, The University of Queensland, Brisbane, Australia
[7] Arboretum Bajnem, Institut National Recherche Forestiere, Alger, Algeria

## ABSTRACT

Prolonged diapause occurs in a number of insects and is interpreted as a way to evade adverse conditions. The winter pine processionary moths (*Thaumetopoea pityocampa* and *Th. wilkinsoni*) are important pests of pines and cedars in the Mediterranean region. They are typically univoltine, with larvae feeding across the winter, pupating in spring in the soil and emerging as adults in summer. Pupae may, however, enter a prolonged diapause with adults emerging one or more years later. We tested the effect of variation in winter temperature on the incidence of prolonged diapause, using a total of 64 individual datasets related to insect cohorts over the period 1964–2015 for 36 sites in seven countries, covering most of the geographic range of both species. We found high variation in prolonged diapause incidence over their ranges. At both lower and upper ends of the thermal range in winter, prolonged diapause tended to be higher than at intermediate temperatures. Prolonged diapause may represent a risk-spreading strategy to mitigate climate uncertainty, although it may increase individual mortality because of a longer exposure to mortality factors such as predation, parasitism, diseases or energy depletion. Climate change, and in particular the increase of winter temperature, may reduce the incidence of prolonged diapause in colder regions whereas it may increase it in warmer ones, with consequences for population dynamics.

# INTRODUCTION

Although diapause is recognized as a common strategy among insects to overcome unfavourable periods, few studies have focused on diapause spanning more than one year (*Danks, 1987*; *Soula & Menu, 2005*). In general, diapause represents a break in development maximising the chance to survive predictably adverse conditions (*Tauber, Tauber & Masaki, 1986*). By contrast, the expression of prolonged diapause varies within a population

Corresponding author
Andrea Battisti,
andrea.battisti@unipd.it

and is generally considered a way to spread the risk due to variably unfavourable conditions amongst years (*Menu, Roebuck & Viala, 2000*). If organisms can use an external or internal signal to predict unfavourable conditions, which reduce fitness, phenotypic plasticity can intervene to minimize the costs (*Pigliucci, 2001*). In contrast, if unfavourable conditions cannot be predicted by any cues, their negative effect on fitness can be mitigated using a "risk-spreading" (*Den Boer, 1968*), or "bet hedging" strategy (*Slatkin, 1974*). Prolonged diapause is a common example of risk-spreading in insects, as it creates heterogeneity within populations by extending the life cycle of a fraction of individuals which stay in a resistance form regardless of external conditions. In an environment with uncertain fluctuations, this strategy can be beneficial in the long-term because it mitigates the worst-case scenario of massive mortality when all offspring are exposed to unpredicted stress. The added heterogeneity reduces between-individual variance in fitness in case of stress by increasing the geometric mean fitness in the long run (*Seger & Brockmann, 1987*; *Starrfelt & Kokko, 2012*). Individuals undergoing prolonged diapause may, however, suffer increased mortality (*Sims, 1983*) or reduced performance (*Matsuo, 2006*) compared to their counterparts, because they spend energy (*Hahn & Denlinger, 2007*) and are exposed to adverse factors and natural enemies for longer durations.

If resource availability or some other essential environmental factors show a strong multiannual cyclic component, a large part of the individuals in a population may enter prolonged diapause, such as in yucca moth depending on inflorescence availability in desert areas (*Powell, 1974*), or in cone insects depending on bumper crop years in boreal forests (*Hanski, 1988*). More irregular conditions may lead to more variation in the expression of prolonged diapause within the same population because the variance balances the fitness costs and benefits of the extended cycle when risks are uncertain. Whether the diapause attitude can be inherited has been discussed in the literature, but there is no clear data available (*Tauber & Tauber, 1981*; *French, Coates & Sappington, 2014*). Maternal effects have been considered, especially in relation to abiotic (climate) and biotic (nutrition, diseases) conditions experienced by the parents (*Mousseau & Dingle, 1991*; *Denlinger, 2002*). Overall, climate plays a major role in all ectotherm life-histories and affects populations in a way that is generally independent of density, whereas biotic factors are strictly linked to population density and can thus be more predictable (*Hanski, 1988*).

In Europe, Middle East, and Northern Africa, the winter pine processionary moths are represented by two sister species, *Thaumetopoea pityocampa* and *Th. wilkinsoni* (Lepidoptera, Notodontidae), which are pests of conifers and a threat to human and animal health because of urticating setae (*Roques, 2015*). Larvae feed during winter and generally adopt a univoltine life cycle, with the exception of Corsica where *Th. pityocampa* is strictly semivoltine (*Geri, 1983*). Temperature influences the duration of both larval and pupal stages (*Démolin, 1969*; *Berardi et al., 2015*; *Robinet, Laparie & Rousselet, 2015*). After completing their development in silk tents, larvae leave the host tree in a typical head-to-tail procession in search of a suitable pupation site in the soil. Burying 5–20 cm into the soil, they spin a cocoon and enter a phase of prepupal diapause, spanning 20 to 50 days (*Salman et al., 2018*). Breaking their prepupal diapause, univoltine individuals turn into pupae and enter a pupal diapause ranging from 1 month at high elevations/latitudes to 5 months at

low elevations or latitudes. Such a difference in duration has been interpreted as a way to optimize the larval development in the next generation (*Démolin, 1969*; *Robinet, Laparie & Rousselet, 2015*). The duration of this pupal diapause appears genetically determined, as populations from different elevations or latitudes reared under the same conditions over a few generations maintain their typical phenology (*Petrucco-Toffolo et al., 2018*). Moths generally emerge in summer, but a variable fraction of the individuals may postpone their emergence and enter a prolonged pupal diapause that may last, as far as we know, up to 9 years in *Th. wilkinsoni* (*Halperin, 1990*) and 8 years in *Th. pityocampa* (*Salman et al., 2016*).

The mechanisms inducing and maintaining prolonged diapause in the winter pine processionary moths are unknown and do not appear to be genetically determined like normal pupal diapause. However, a temperature-based hypothesis suggests that prolonged diapause in the pine processionary moths is a developmental strategy to cope with adverse temperature conditions, at both ends of the scale (*Démolin, 1969*). It suggests that the onset of summer is too hot for the normal development of eggs and young larvae at the lower/southern edge of the range, whereas the winter is too cold for further larval development at the upper/northern edge (*Démolin, 1969*; *Robinet, Laparie & Rousselet, 2015*). Soil moisture (*Torres-Muros, Hódar & Zamora, 2017*), food quality or availability (*Battisti, 1988*), population density, and natural enemies have been proposed as additional factors influencing the expression of prolonged diapause (*Geri, 1983*), but remain to be experimentally demonstrated. *Battisti et al. (2005)* identified the temperature of the so-called 'cold period' (December-January-February) as a key predictor for larval survival, showing that climate change at both high elevation and latitude close to the range edges can trigger the successful establishment beyond the past range and cause the current expansion of this pest.

Here, to test the temperature hypothesis proposed by *Démolin (1969)*, we collected and analysed all the available evidence of prolonged diapause across the range of *Th. pityocampa* and *Th. wilkinsoni*, taking into account the genetic structure of the group (*Kerdelhué et al., 2009*; *El Mokhefi et al., 2016*). Populations of *Th. pityocampa* from Corsica were excluded because of their strict semivoltinism (*Geri, 1983*). We specifically tested the effect of winter temperature as a driver of prolonged diapause in these species, and the costs of prolonged diapause in terms of mortality. We finally discuss the role of climate change on prolonged diapause and the potential effects on the population dynamics of these important pests.

## MATERIAL AND METHODS
### Data collection
Published and unpublished data (initial available documents, $n = 42$) were retrieved from scientific databases and from institutional reports. Because we hypothesized winter temperature to be a mechanism causing prolonged diapause, we only selected studies that met the following criteria: (i) the total number of individuals is given, (ii) insects are collected using one or more of the five methods described below, (iii) the number of moths emerged in the year of collection and in the subsequent year(s) is given, (iv) the number of
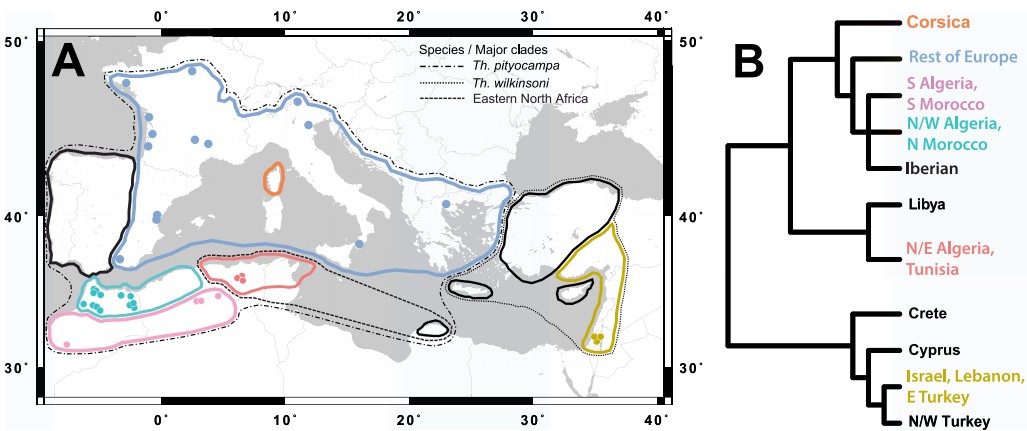

**Figure 1  Location of the study sites in relation to the distribution of the pine processionary moth.** Sites used for the analysis of prolonged diapause at the pupal stage, indicated with coloured dots within each species and subclade of the pine processionary moth (A), and tree of genetic structure based on *Kerdelhué et al. (2009)* and *El Mokhefi et al. (2016)* (B). Data were not available for the subclades indicated in black.

**Table 1  Summary list of the study sites.** Distribution of the 37 study sites among genetic clades and subclades of pine processionary moths according to *Kerdelhué et al. (2009)* and *El Mokhefi et al. (2016)*. The time period of sampling and the pine host species are also given.

| Clade | Subclade | No. of sites | Years | Host plant |
|---|---|---|---|---|
| Th. pityocampa | Rest of Europe | 14 | 1971–2015 | *Pinus brutia* *P. halepensis* *P. nigra* *P. pinaster* |
| Th. pityocampa | N/W Algeria, N Morocco | 13 | 1988 | *P. halepensis* |
| Th. pityocampa | S Algeria, S Morocco | 3 | 1988–1995 | *Cedrus atlantica* *P. halepensis* |
| Eastern North Africa | N/E Algeria, Tunisia | 4 | 1983 | *P. halepensis* |
| Th. wilkinsoni | Israel, Lebanon, E Turkey | 3 | 1964–1970 | *P. halepensis* |

dead larvae/pupae is given for a specific year of sampling (hereafter referred to as a cohort), and (v) the average air temperature of the cold period (December-January-February) of the studied years is given. Information was obtained for seven countries from a total of 13 documents and 36 sites in the period 1964–2015 (Fig. 1, Table 1, Data S1). These sites belong to 6 out of 11 genetic subclades of the pine processionary moths described by *Kerdelhué et al. (2009)* and *El Mokhefi et al. (2016)* (Fig. 1) and cover most of the climatic range of the species (*Roques, 2015*). The data span latitudes from 31.3°N to 47.8°N, longitudes from −7.9° to 35.2° (Fig. 1), and elevations from 10 to 1,910 m. For 11 sites, more than one year (up to six) of observations was available, giving a total of 64 individual datasets on the occurrence of prolonged diapause.

For the year(s) of sampling, the temperature of each site was collected from the source document ($n = 41$) or from records of the nearest weather station ($n = 23$) (Data S3). Average monthly temperature of the cold period (December-January-February) (*Battisti et al., 2005*; *Roques, 2015*) was available for all datasets and was included in the analyses. For those sites with both average and minimum temperature, the correlation between the two predictors was very strong ($R^2 = 0.94$), indicating that the choice of either metric would only marginally affect the results.

The mortality percentage was calculated as the number of individuals that did not produce moths with respect to the initial number of individuals in the cohort. The prolonged diapause percentage was calculated as the number of individuals that emerged over the year(s) following the year of pupation, or that were still alive when the experiment was over one or more years after pupation (emerged after prolonged diapause + living pupae at the last check), with respect to the total number of emerged or living individuals (emerged univoltine + emerged after prolonged diapause + living pupae at the last check).

Five collection methods were used: (i) digging out pupae from natural pupation sites previously marked by following mature larvae during their processions until they started to bury themselves; (ii) collecting mature larvae during their pupation processions on the ground before they bury and placing them in pots with substrate to stimulate pupation; (iii) intercepting mature larvae at the beginning of their procession on tree trunks with collar traps during their way down to the ground, and placing them in mesh-capped pots or geotextile bags with substrate to stimulate pupation, on site or at the laboratory; (iv) collecting tents full of mature larvae ready for processions, placing them in outdoor cages or mesh-capped pots with substrate, separately or clustered, until pupation; (v) collecting tents with 4th or 5th instar larvae, placing them in outdoor cages or mesh-capped pots with substrate, separately or clustered, and supplying fresh pine foliage *ad libitum* until pupation. The year of pupation was known for each colony and it was then possible to discriminate univoltine adults emerging in the same year from those emerging in the following year(s) after prolonged diapause. Insects collected using methods (i) and (iii) were individually placed in glass vials in the laboratory for future monitoring as soon as they pupated in the field, while those collected using method (ii) were placed in vials only after the univoltine adults emerged. Individuals collected using methods (iv) and (v) and placed in field cages were left and monitored there, while those placed in pots were either left in the substrate or transferred to glass vials after univoltine adults emerged. The monitoring was carried out until no emergence was observed for three consecutive years.

## Data analysis

To test the effect of temperature on prolonged diapause incidence, we used a linear mixed-effect model (Data S2). The model included the linear, quadratic and cubic terms of temperature as fixed effects and site nested within genetic subclade as random effect. We omitted one data point using Cook's distance (value > 5). This very influential point was related to the maximum observed mean temperature. The consideration of subclade and site as random effects accounted for the biological and spatial dependence in the data. It is important to stress that our model tested primarily the effect of spatial variation in winter

temperature across the species' ranges, and only secondarily the inter-annual variation in temperature within the sites. The analyses were run using the *nlme* package in R (Data S3) (*R Core Team, 2016*). A model with the same random structure was used to test the association between the incidence of prolonged diapause and pupal mortality.

## RESULTS

Prolonged diapause occurred in all genetic subclades of *Th. pityocampa* and *Th. wilkinsoni* for which data were available (Fig. 2A). All 37 sites showed at least one year with occurrence of prolonged diapause, and only five out of 65 individual datasets or cohorts did not show prolonged diapause. The incidence of prolonged diapause in the first year after pupation varied greatly in time and space. The mixed model indicated that temperature affected the incidence of prolonged diapause in a non-linear fashion, i.e., linear ($t_{1,25} = -5.481$, $P < 0.001$), quadratic ($t_{1,25} = -3.687$, $p = 0.001$) and cubic terms ($t_{1,25} = -2.395$, $p = 0.025$) were significant. Both cold winters (temperature of December-January-February around 0 °C) and warm winters (above 10 °C) resulted in a high incidence of prolonged diapause, whereas intermediate winters (between 2 and 10 °C) resulted in less frequent prolonged diapause, irrespective of subclades (Fig. 2A). However, the effect of warm winters was less pronounced than that of cold winters. The five datasets with no occurrence of prolonged diapause and almost all those with less than 25% of prolonged diapause were observed at temperatures between 2 and 10 °C (Fig. 2A).

Pupal mortality was positively and significantly correlated to prolonged diapause incidence (Fig. 2B). Mortality varied greatly in time and space. It reached the highest values at both cold (around 0 °C) and warm (above 10 °C) winter sites, indicating that individuals in cohorts showing a high incidence of prolonged diapause are more likely to die than those showing a low incidence. Mortality rates lower than 25% were associated with low incidence of prolonged diapause, especially in the subclade *Rest of Europe*. The mortality factors were not systematically addressed in each study, so it was not possible to compare their relative effects.

## DISCUSSION

Winter temperature experienced by larvae was found to be a reliable predictor of the prolonged diapause incidence in the winter pine processionary moths pupae: prolonged diapause tends to increase at both lower and upper ends of the thermal range in winter, although the incidence remained highly variable among samples. The adaptive value of prolonged diapause in this species can be seen at both ends of the temperature range, presumably for different reasons. In colder sites and years, the harsh winter challenges larval survival that depends essentially on reaching thermal feeding thresholds (*Battisti et al., 2005*). Prolonged diapause in this case dilutes the risk of being exposed again to long starvation associated with unusually cold weather among years (*Danks, 1987*; *Battisti et al., 2005*). In warmer sites and years, peaking summer temperatures are assumed to overreach upper thermal limits of eggs and young larvae and impair their survival (*Halperin, 1990*; *Santos et al., 2011*). In this case winter temperature works as a proxy of another thermal

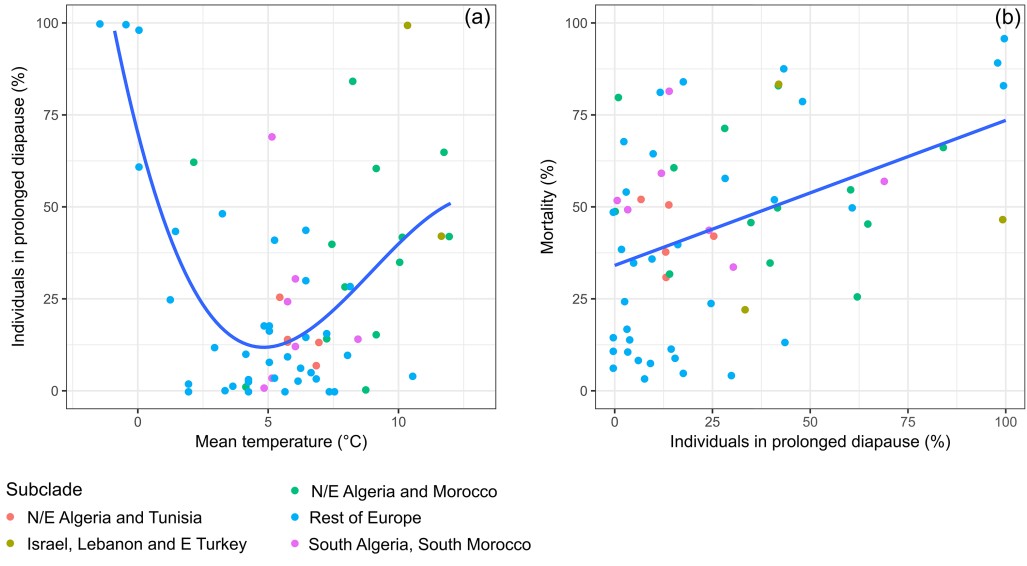

**Figure 2** **Variation of prolonged diapause rate in relation to winter temperature and relationships between prolonged diapause and pupal mortality.** (A) Relationship between the daily mean temperature during the cold period of the larval development (December, January, and February) and the percentage of prolonged diapause in the cohorts of pupae originating from larvae experiencing those temperatures. The fitted line represents predictions of a mixed model including subclade and site as random effects (Diapause = $80.39 - 28.69$ Temp + $3.96$ Temp$^2$ $-0.15$ Temp$^3$, all terms $P < 0.05$, $n = 64$). (B) Relationship between the percentage of individuals entering prolonged diapause and the pupal mortality observed in the first year of prolonged diapause. The fitted line represents a prediction of a mixed model including subclade and site as random effects (Mortality = $37.88 + 0.364$ Diapause, $P < 0.001$, $R^2 = 0.18$, $n = 62$).

stressor that could become limiting later in the year when moths would emerge and lay eggs. Thus, entering prolonged diapause at the southern edge of the range where both winter and summer tend to be warmer would allow spreading the risk through time and facilitate the persistence of populations (*Hanski, 1988*). Slight thermal changes may drastically influence the survival rate in populations close to their biological limits. This is particularly problematic at the edge of the distribution range (*Renault et al., 2018*). In such conditions where the thermal margin is low, weather conditions may exceed biological limits any year. Therefore, prolonged diapause in a fraction of individuals, even at a moderate temperature generally deemed harmless for growth and survival, may often be the safest bet. This strategy differs from that of other Lepidoptera that use a fixed prolonged diapause to synchronize with the availability of resources, such as the yucca inflorescences for yucca moth (*Powell, 1974*) and the spruce seed cones for the seed cone moth (*Hanski, 1988*).

Variance in prolonged diapause duration could not be tested in this study because most reports did not survey the emergence of the moths for more than three years. As durations up to 8–9 years have been found at a few sites (*Halperin, 1990*; *Salman et al., 2016*), an hypothesis is that the intensity of the winter stress could be proportional to the duration of prolonged diapause, causing a higher variance and a longer dilution of the risk in the most severe conditions. In addition, high density of larvae would result

in higher incidence of prolonged diapause, as damaged trees may be unpalatable for the larvae of the following generations (*Battisti, 1988*). The higher the density of larvae, the higher the unpalatability of needles, the longer would be the diapause (*Salman et al., 2016*). This sort of density-dependent prolonged diapause could combine with mechanisms induced by abiotic stresses, e.g., temperature. Unfortunately, many studies on prolonged diapause examined in the present work did not report population density at the time of sampling. Other putative factors (such as epidemics, predation and parasitism) may respond to density variation, with or without lag effects, and thus prolonged diapause can be hypothesized as an escape in time from such biotic lethal factors (*Hanski, 1988*). Numerous parasitoids and predators of the pine processionary moth are univoltine and cannot track and keep synchronization with individuals entering prolonged diapause (*Battisti, Bernardi & Ghiraldo, 2000*).

Irrespective of the thermal conditions, pupal mortality increased with prolonged diapause incidence likely because of a longer exposure to chronic mortality factors (*Sims, 1983*), including natural enemies or depletion of energy reserves (*Hahn & Denlinger, 2011*). These higher mortality risks associated with prolonged diapause are expected and important but they do not contradict the benefits of prolonged diapause as a diversifying risk-spreading strategy. In fact, more attention should be given to climatically non-typical years when benefits of such a strategy would be expected, ultimately securing the survival of part of the offspring and long-term persistence of populations (*Menu, Roebuck & Viala, 2000*). Prolonged diapause could also limit the risks of local extinction after the generally high mortality observed at outbreak density (*Tamburini et al., 2013*; *Li et al., 2015*; *Salman et al., 2016*). Survival, rather than reproduction, is more important in a declining population or a population close to its ecological limits, and is secured by entering prolonged diapause. However, whether increased frequency of prolonged diapause contributes to declining population growth rates or the other way around, is still not clear (*Hanski, 1988*). In addition, the phenology of the winter pine processionary moths is severely constrained by climatic factors (*Robinet, Laparie & Rousselet, 2015*), and it could be that in certain years the required physiological conditions to complete univoltine development are not met, thus leaving no alternative to prolonged diapause that eventually may become fixed as in Corsica mountains (*Geri, 1983*).

Climate change may affect prolonged diapause in different ways. As winter is experienced in two different life stages, the growing larva and the pupa undergoing prolonged diapause, it is likely that their respective vulnerability to stressful conditions differ. Larvae feeding on trees are exposed to low, fluctuating temperatures, but are capable of selecting suitable microclimates and have a silk tent helping to regulate temperature (*Battisti et al., 2005*), while pupae in the soil are passively buffered against temperature extremes. In addition, both stages have complex interactions with precipitation and moisture, and related pathogenic agents (*Torres-Muros, Hódar & Zamora, 2017*). Observations carried out between 1999 and 2016 in the coldest edge of the species' geographical range in the Alps indicated that prolonged diapause incidence markedly decreased as a consequence of climate warming (*Salman et al., 2016*). This observation is congruent with our finding that prolonged diapause likely mitigates the risk associated with low winter temperatures in

cold areas of the distribution, since winter warming lowers the climatic barrier. Likewise, future studies should test whether climate warming in warm areas of the range intensifies the risks associated with summer heats and increase the incidence to prolonged diapause, as predicted by our results. Such changes in the incidence of prolonged diapause, and related mortality, have to be taken into consideration when modelling population dynamics and range expansion, since prolonged diapause can alter the long-term success of a population in a given area (*Li et al., 2015*). For example, a reduction of prolonged diapause incidence would make predictions easier because it will emphasize the role of density-dependent factors, which are important in this species (*Tamburini et al., 2013*; *Toïgo et al., 2017*).

## CONCLUSION

Winter temperature variation across both species range is an important extrinsic factor for geographic variation in prolonged diapause incidence. While prolonged diapause was minimal in areas where winters are intermediate, both low and high winter temperatures are conducive to high frequency of prolonged diapause in pupae, likely because of different avoidance mechanisms of lower and upper biological limits in different insect life stages. However, a number of other factors (such as genetic determination, parasitism, predation, diseases, intraspecific competition) have to be considered for a better understanding of the physiological mechanism regulating prolonged diapause in winter pine processionary moths. In particular, population density and nutritional status of larvae before pupation may reveal the role of prolonged diapause in the population dynamics of this important defoliating pest.

## ACKNOWLEDGEMENTS

The authors warmly acknowledge Philipp Lehmann for comments on an earlier version of the manuscript and Paolo Paolucci for drawing the figures. Alexis Bernard and Patrick Pineau helped in the collection of data from a few sites in France. Anonymous reviewers are also warmly acknowledged.

### Funding

Work supported by Fondazione Cassa di Risparmio di Padova e Rovigo to Md H.R. Salman, University of Padova to Myron P. Zalucki and Folco Giomi, Spanish Ministry of the Environment (PROPINOL PN22/2008 and CONSOLIDER-MONTES CSD2008-00040) to José A. Hodar, DIAMETABO project of the INRA EFPA department to Mathieu Laparie, European Union's Horizon 2020 research and innovation programme under grant agreement N. 771271 HOMED (Holistic Management of Emerging forest pests and Diseases) to Andrea Battisti. The funders had no role in study design, data collection and analysis, decision to publish, or preparation of the manuscript.

## Grant Disclosures

The following grant information was disclosed by the authors:

Fondazione Cassa di Risparmio di Padova e Rovigo.

University of Padova.

Spanish Ministry of the Environment: PROPINOL PN22/2008, CONSOLIDER-MONTES CSD2008-00040.

DIAMETABO project of the INRA EFPA department.

European Union's Horizon 2020 research and innovation programme.

## Competing Interests

The authors declare there are no competing interests.

## Author Contributions

- Md H.R. Salman performed the experiments, analyzed the data, authored or reviewed drafts of the paper, approved the final draft.
- Carmelo P. Bonsignore and Ahmed El Alaoui El Fels contributed reagents/materials/-analysis tools, approved the final draft.
- Folco Giomi approved the final draft, data discussion and interpretation.
- José A. Hodar contributed reagents/materials/analysis tools, approved the final draft.
- Mathieu Laparie analyzed the data, contributed reagents/materials/analysis tools, authored or reviewed drafts of the paper, approved the final draft.
- Lorenzo Marini analyzed the data, prepared figures and/or tables, approved the final draft.
- Cécile Merel performed the experiments, approved the final draft.
- Myron P. Zalucki authored or reviewed drafts of the paper, approved the final draft.
- Mohamed Zamoum contributed reagents/materials/analysis tools, approved the final draft.
- Andrea Battisti conceived and designed the experiments, performed the experiments, analyzed the data, prepared figures and/or tables, authored or reviewed drafts of the paper, approved the final draft.

## Data Availability

The raw data and codes are available as Supplementary File.

## Supplemental Information

Supplemental information for this article can be found online at http://dx.doi.org/10.7717/peerj.6530#supplemental-information.

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
