# Peer review of "Winter temperature predicts prolonged diapause in pine processionary moth species across their geographic range"

_PeerJ, doi:10.7717/peerj.6530_

## Round 0.1 · original submission · Major Revisions

Dear Andrea,

Two detailed and comprehensive reviews have been provided by leaders in entomological research.

Please consider improved reporting and presentation of statistical analyses undertaken, and the context for the studies drawn upon. There are specific alternative statistical analyses suggested by Reviewer 2, please consider these with direct consultation with a statistical advisor and provide response.

The introduction requires greater depth and detail as both reviewers suggest. Please reflect upon the influence of both inter-annual temperature as well as geographical location on your findings.

Regards
Steve

·

Basic reporting

The paper is well structured (standard academic format) and provides new interesting results but requires more explanation in each section for the reader to fully grasp the research. References are well cited and the figures are informative and high quality.

Introduction:

I find the abbreviation of PD distracting and would prefer the authors just write prolonged diapause.

LN 49. “and is generally considered as a way to spread the risk” this sentence makes PD sound like more of a deliberate choice than it is and implies there are other explanations. Could it just be a fluke or mistake rather than a strategy? For readers with little background knowledge of PD it would help to have a little more background.

The intro does not provide the reader with enough background on the life history of PPM. When does it pupate? For how long? What mechanisms cause PD and how do they cause it? Without a thorough understanding of these things one cannot follow the paper or results.

ln 60. what is facultative PD? why does it occur? does this mean all PPM in this range have PD?

ln67. What is the temperature hypothesis? Is this a formal hypothesis established by Demolin? Did the authors just invent it? Describe this hypothesis if it is an established hypothesis in the literature.

Experimental design

The research is original and unique. They take advantage of collaborations with many authors to combine data from a large geographic range.

ln79-83. I do not know why these criteria were chosen since the life cycle is not explained or the proposed mechanisms for PD. This section would benefit from a statement like: Since PPM pupate during this time of year and we predict x to be a mechanism causing PD we only selected studies that include the following criteria….

It would also be helpful to know how the experiments were conducted (briefly). Were larvae collected from trees then reared in the lab? were they kept outside? how can one track a wild cohort for multiple years? I can’t judge the soundness of the criteria without more knowledge of the studies and the PPM.

ln90. Why the average temperature? I thought ppm was affected by minimum temperatures? Many insects are more sensitive to extremes than averages. Please explain this choice.

ln102. How do the author's account for latitude? do they need to? In the introduction they say PD is common or facultative in the northern and southern edges of the range.

Validity of the findings

Results
Would the relationship between temperature and PD or survival be the same if replaced with latitude? For example would figure 2a look the same if the x axis was latitude instead of temperature? From the map it doesn’t look like it but showing this might strengthen the argument that it is temperature and occurs anywhere within the range that temperatures reach the specified lows or highs.

There is no report of the statistics related to clade. This was included in the analyses (methods ln104) but we do not see the results of how much clade influenced PD compared to temperature.

Actually I do not see any statistics reported accept in the figure legends. These should also be in the text.

Discussion
ln148. Again it would be helpful to know how the studies were conducted. In addition I do not understand from this sentence how PD can be assessed from a one year study?
This paragraph should expand on potential mechanisms. For example heat is a stressor but how does this translate physiologically into PD.

The explanations that exposure to heat or cold stress leads to PD which reduces encounters with those stressors the next year is not intuitive. It implies to me that every hot year is followed by another hot year that the animal would want to avoid. I would think an exceptionally hot or cold year is more likely to be followed by an average year. In this case the risk of PD does not make sense. Expand this line of discussion.

Elsewhere like ln152 the authors imply poor nutrition could lead to PD. Is this via the same stress mechanism as cold or a different one. Someone without a vast knowledge of diapause would not be able to follow this.

ln187. Telling us different avoidance mechanisms are important without discussing what they may be is not enlightening and does not allow the reader to ponder hypotheses about this or other species and expand on and cite the work in the future.

Additional comments

The research presented is sound and important but the manuscript needs revision to help the reader get the most from the results and to provide proper context for the results.

Reviewer 2 ·

Basic reporting

I have read with great interest the manuscript (#28902) “Winter temperature predicts prolonged diapause in pine processionary moth across its geographic range” by Salman et al. and the supporting material. The English language is good and understandable, and as I am not a native English speaker, I have no further linguistic comments. Also the Figures are informative and relevant, high quality, and well described. Raw data is (partially) supplied in the electronic supporting material.
The Introduction gives a rather short overview of prolonged diapause, focusing on the role of adverse abiotic factors (winter temperatures) as the selective or triggering factor behind it (lines 61-66) in the Pine Processionary Moth (PPM); in the Discussion (lines 151-154) the potential role of food shortage is shortly mentioned, but I think that the avoidance of parasitoids and diseases should also be discussed. Overall, I think that the paper would benefit from a more thorough introduction to and discussion of the theories of prolonged diapause (to better relate their empirical study to the existing theory); I think that the authors should read and cite the article by Hanski (1988) Four kinds of extra long diapause in insects: A review of theory and observations. Ann. Zool. Fennici 25, 37-53.

Experimental design

The empirical data base of this study consists of published and unpublished studies done during an extended time frame (from 1971 to 2012) in different parts of the distribution area of PPM. The hypothesis tested in the present manuscript has probably not been the main question addressed in these studies, and the data is very heterogeneous (as detailed in the supporting material): both field and field/lab studies, different protocols, etc. Also, it has not been explicitly stated with each data whether the temperatures used are air or soil temperatures; the often repeated statement “Retrieved from the documents” is not informative/accurate enough.
The authors write (lines 106-108) that “It is important to stress that our model tested the effect of spatial variation across the species range, and not the inter-annual variation in temperature within the sites.” However, if I have understood correctly their data (given in supporting material), there are also sites with several observation years and with inter-annual variation in temperatures (e.g., six years for Vielle-St-Girons, France). So, I think that the effects of both inter-annual temperature variation and location effects are studied in this manuscript.

Validity of the findings

The authors state (line 103) that they have “used a linear mixed-effect model”. I think this is not a correct expression, as their model has also a quadratic (non-linear) term. Furthermore, I think that the selection of a polynomial model with only two terms (linear and quadratic) does not use all the information in the data, and leads to (partially) wrong conclusions. On the basis of a visual inspection of Figure 2(a), I think the data should be analyzed as a series of polynomial mixed-effect models with first, second, third and (maybe) fourth order terms, and the performance of these alternative models should be compared with information based criteria (e.g. AICc). Another option would be mixed model ANOVA comparing the groups “< +2oC”, “2-10oC”, and “>10oC”. I don’t agree with the conclusion of the authors that “temperatures .. .. higher than 10oC were associated with higher PD incidence than cohorts exposed to intermediate temperatures”. It seems to me, that ANOVA would not reveal a significant difference in the PD between intermediate temperatures [ “2-10oC”] and high temperature [ “>10oC”] due to the very high variation within the latter group [PDmin = 4.2%, PDmax = 99.6%].

Additional comments

Please, find the above cited article by Hanski (1988) here:
http://www.sekj.org/PDF/anzf25/anz25-037-053.pdf

---

## Round 0.2 · Major Revisions

Dear authors, one of the previous reviewers and a new reviewer has assessed the revised manuscript as well as myself.

Many of the original comments and suggestions have been addressed in this revision, thank you for doing this. However, the language and quality of English in these revised sections are not adequate and has led to greater confusion and dissatisfaction when reading through. Please address this as a priority - the level of English language must be consistent and of a high quality.

Please can you also address a comment from reviewer 3, who suggested reject based on the poor quality of communication when reading through the revised version. Within the manuscript it is necessary to discuss the point R3 raises that there is only one site out of the 46 where temperatures reach below 0 freezing. Also, please discuss when investigating prolonged diapause, the issues that many of the 46 studies only provide one year of data. This will provide a more thoughtful and honest appraisal of the strength and weakness of this novel and valuable data analysis approach you have undertaken.

The recommendation remains Major Revision until these issues can be resolved

Regards
Steve

·

Basic reporting

Basic reporting is OK but as I describe in my review there are many places where text and ideas are confusing and need to be clarified.

Experimental design

Excellent and novel.

Validity of the findings

Data and statistics sound. The authors made changes that help us understand how data were collected. Some aspects of concluding paragraph and discussion can be clarified. Suggestions in review document.

Additional comments

Many good changes made in response to previous reviews were completed. However, some of the new text is confusing and needs rewritten.

Reviewer 3 ·

Basic reporting

No comment

Experimental design

No comment

Validity of the findings

No comment

Additional comments

I read the submitted manuscript, the comments of the reviewers, the response of the authors and the modified manuscript. The new version was improved a lot compared with the original version, however I still found manuscript confusing and some of the fair criticisms of the reviewers are no solved in the new version.
The paper deals with the role of the winter temperatures to predict prolonged diapause in Pine processionary moth. Most of the data are from published papers with heterogeneous questions and experiments. Authors claim that the study range from 1964 to 2015, but in many of the sites the data are from one year, not enough to study prolonged diapause.
With the data that they use they cannot reject other factors that may influence prolonged diapause as limited food resources or parasitism avoidance
The main conclusion is that insect cohorts exposed to average winter temperatures lower than 0ºC were associated with prolonged diapause, but according with data provided in Supplemental Data set 2 there is a single site in the study (of the 46 with data of winter temperatures: Venosta), that reach T below 0 during winter. And, if I understand well the Supplemental data set 2, even in that site diapause in one year was as low as 2.1 (I guess %).
In my opinion the paper address an important question, but the data that they use is not the best option to solve it.

---

## Round 0.3 · accepted · Accept

Well done on a comprehensive response to the second round of revisions and comments from two reviewers.

The communication in the intro and discussion is greatly improved and I accept and understand your response to R3 questions regarding number of years of data and temperature range not extending beyond zero.

I am happy to make an 'accept' recommendation without further need for expert review.

Regards
Steve

#